# Global assessment of existing HIV and key population stigma indicators: A data mapping exercise to inform country-level stigma measurement

**Carrie Lyons** [1]*, **Victoria Bendaud**[2], **Christine Bourey** [3], **Taavi Erkkola**[2], **Ishwarya Ravichandran**[1], **Omar Syarif**[4], **Anne Stangl** [5,6], **Judy Chang** [7], **Laura Ferguson** [8], **Laura Nyblade** [9], **Joseph Amon** [10], **Alexandrina Iovita** [11], **Eglė Janušonytė** [12,13], **Pim Looze**[4], **Laurel Sprague** [2], **Keith Sabin** [2], **UNAIDS Task Team**¶, **Stefan Baral** [1], **Sarah M. Murray** [3]

1 Epidemiology Department, Johns Hopkins Bloomberg School of Public Health, Baltimore, Maryland, United States of America, 2 UNAIDS, Geneva, Switzerland, 3 Mental Health Department, Johns Hopkins Bloomberg School of Public Health, Baltimore, Maryland, United States of America, 4 Global Network of People Living with HIV (GNP+), Amsterdam, the Netherlands, 5 International Health Department, Johns Hopkins Bloomberg School of Public Health, Baltimore, Maryland, United States of America, 6 Hera Solutions, Baltimore, Maryland, United States of America, 7 International Network of People Who Use Drugs (INPUD) Secretariat, London, United Kingdom, 8 Keck School of Medicine, USC Institute on Inequalities in Global Health, University of Southern California, Los Angeles, California, United States of America, 9 Global Health Division, International Development Group, RTI International, Washington, DC, United States of America, 10 Dornsife School of Public Health, Drexel University, Philadelphia, Pennsylvania, United States of America, 11 Communities, Rights and Gender Department, The Global Fund to Fight AIDS, Tuberculosis and Malaria, Geneva, Switzerland, 12 International Federation of Medical Students' Associations, Copenhagen, Denmark, 13 Faculty of Medicine, Vilnius University, Vilnius, Lithuania

¶ Membership of the UNAIDS Task Team is listed in the Acknowledgments.
* clyons8@jhu.edu

**Data Availability Statement:** This data mapping exercise used several data sources, which are outlined below. 1. UNAIDS National Commitments

## Abstract

### Background

Stigma is an established barrier to the provision and uptake of HIV prevention, diagnostic, and treatment services. Despite consensus on the importance of addressing stigma, there are currently no country-level summary measures to characterize stigma and track progress in reducing stigma around the globe. This data mapping exercise aimed to assess the potential for existing data to be used to summarize and track stigma, including discrimination, related to HIV status, or key population membership at the country level.

### Methods and findings

This study assessed existing indicators of stigma related to living with HIV or belonging to 1 of 4 key populations including gay men and other men who have sex with men, sex workers, people who use drugs, and transgender persons. UNAIDS Strategic Information Department led an initial drafting of possible domains, subdomains, and indicators, and a 3-week e-consultation was held to provide feedback. From the e-consultation, 44 indicators were proposed for HIV stigma; 14 for sexual minority stigma (including sexual behavior or

and Policies Instrument (NCPI) database[1]. The NCPI Data Platform is the online platform for policy data related to the AIDS response, and the database contains country-submitted responses to selected indicators from the NCPI questionnaire. NCPI data are publicly available, and datasets can be downloaded from the UNAIDS databases on the website. 2. Multiple Indicator Cluster Surveys (MICS)[2]. MICS data are publicly available, and datasets can be downloaded from the UNICEF webpage. Reports of MICS surveys can be directly downloaded from the website. Data are available upon request to MICS by registering as a MICS data user. 3. Demographic and Health Surveys (DHS)[3]. Survey data are publicly available and can be downloaded directly from the website. DHS surveys can be pulled from StatCompiler and individual country DHS reports are also available for review. 4. People Living with HIV Stigma Index 1.0 surveys[4]. Country reports detailing results of the People Living with HIV Stigma Index 1.0 survey are available by request to plhivstigmaindex@gnpplus.net. 5. International Lesbian, Gay, Bisexual, Trans and Intersex Association (ILGA)[5]. Data are available on the ILGA website. 6. The Integrated Biological and Behavioral Surveys (IBBS) utilized in this data mapping exercise have been previously published. Summaries of these data can be found in published articles [6, 7] Raw data cannot be shared publicly because of sensitive nature of the responses and the criminalization or other potential social consequences of reported behaviors, identities, or experiences. Data are available from the Johns Hopkins Key Populations Program and Center for Implementation Research (contact via CenterforImplementationResearch@live.johnshopkins.edu) for researchers who meet the criteria for access to confidential data.

**Funding:** This research was funded through a consultancy agreement with UNAIDS, titled Developing Indices of HIV and Key Populations-related Stigma and Discrimination (RFQ-2020-03; PR No. 2020/1013928) awarded to the research team consisting of SM, SB, CL. The sponsors contributed to the study design, data access, and are included as co-authors of the manuscript. C.E. L.'s effort was supported by the National Institute of Allergy and Infectious Diseases (NIAID) Johns Hopkins HIV Epidemiology and Prevention Sciences Training Program (5T32AI102623-08). C. B. was supported by the National Institute of Mental Health (NIMH) Johns Hopkins Global Mental Health Training grant (T32MH103210). S.B. was supported by the National Institute of Mental Health, US National Institutes of Health (R01MH110358). Finally, this publication was

orientation) related to men who have sex with men; 12 for sex work stigma; 10 for drug use stigma; and 17 for gender identity stigma related to transgender persons. We conducted a global data mapping exercise to identify and describe the availability and quality of stigma data across countries with the following sources: UNAIDS National Commitments and Policies Instrument (NCPI) database; Multiple Indicator Cluster Surveys (MICS); Demographic and Health Surveys (DHS); People Living with HIV Stigma Index surveys; HIV Key Populations Data Repository; Integrated Biological and Behavioral Surveys (IBBS); and network databases. Data extraction was conducted between August and November 2020. Indicators were evaluated based on the following: if an existing data source could be identified; the number of countries for which data were available for the indicator at present and in the future; variation in the indicator across countries; and considerations of data quality or accuracy. This mapping exercise resulted in the identification of 24 HIV stigma indicators and 10 key population indicators as having potential to be used at present in the creation of valid summary measures of stigma at the country level. These indicators may allow assessment of legal, societal, and behavioral manifestations of stigma across population groups and settings. Study limitations include potential selection bias due to available data sources to the research team and other biases due to the exploratory nature of this data mapping process.

## Conclusions

Based on the current state of data available, several indicators have the potential to characterize the level and nature of stigma affecting people living with HIV and key populations across countries and across time. This exercise revealed challenges for an empirical process reliant on existing data to determine how to weight and best combine indicators into indices. However, results for this study can be combined with participatory processes to inform summary measure development and set data collection priorities going forward.

## Author summary

### Why was this study done?

- Many people living with HIV and key populations, including gay men and other men who have sex with men, sex workers, people who use drugs, and transgender persons, are affected by stigma.

- Stigma is a barrier to HIV prevention, diagnostic, and treatment services and also negatively impacts the health and quality of life of those affected.

- Although reducing stigma is an important pillar of the HIV response, there are currently no established summary measures to characterize the overall level of stigma within a country and track progress in addressing stigma over time.

- The main purpose of this study was to review existing stigma data globally and assess which stigma indicators have the potential to inform a summary measure of stigma at the country level.

made possible by the Johns Hopkins University Center for AIDS Research, an NIH funded program (P30AI094189).

**Competing interests:** The authors have declared that no competing interests exist.

**Abbreviations:** ART, Antiretroviral Therapy; DHS, Demographic and Health Surveys; GNP+, Global Network of People Living with HIV; IBBS, Integrated Biological and Behavioral Surveys; ILGA, International Lesbian, Gay, Bisexual, Trans and Intersex Association; JHSPH-IRB, Johns Hopkins School of Public Health Institutional Review Board; MICS, Multiple Indicator Cluster Surveys; MTAG, Monitoring Technical Advisory Group; NCPI, National Commitments and Policies Instrument; SDG, Sustainable Development Goal.

## What did the researchers do and find?

- In this global mapping exercise, data for stigma indicators detailed in the following sources were reviewed and assessed: UNAIDS National Commitments and Policies Instrument (NCPI) database; Multiple Indicator Cluster Surveys (MICS); Demographic and Health Surveys (DHS); People Living with HIV Stigma Index surveys; HIV Key Populations Data Repository; Integrated Biological and Behavioral Surveys (IBBS); and other databases.

- Through this exercise, 24 HIV stigma indicators and 10 key population indicators were determined to have potential for use in the creation of valid summary measures of stigma at the country level.

## What do these findings mean?

- The results of this study highlight challenges in creating summary measures for stigma experienced by key populations or related to HIV based on existing global data sources.

- However, there is an opportunity to use participatory approaches alongside the existing data to create summary measures to describe and track stigma over time.

## Introduction

Despite significant advancements in HIV prevention, early detection, and treatment, 37.7 million people are living with HIV around the world, and there were an estimated 1.5 million new infections in 2020 [1]. The UNAIDS global targets for 2025 focus on primary prevention of HIV as well as ensuring that 95% of people living with HIV become aware of their status; 95% of people diagnosed with HIV receive Antiretroviral Therapy (ART); and 95% of people living with HIV on ART achieve sustained viral suppression [2]. Collectively, the goal is to end new HIV infections by 2030. However, the stated goal for 2020 was to reduce new infections to 500,000, which was not achieved due in part to limited progress in reducing stigma affecting people at risk for and living with HIV [2]. Stigma has been identified as a social determinant of health and a key barrier to improving health outcomes among people living with HIV [3–8]. As such, stigma continues to present barriers to achieving the HIV prevention and treatment targets by interfering with the provision and uptake of prevention, diagnostic, and treatment services. This is particularly true for key populations (sex workers; gay, bisexual, and other men who have sex with men; people who use drugs; and transgender persons) who may experience stigma relating to actual or assumed HIV status in addition to experiencing intersecting stigma related to their actual or assumed behaviors or identities [2]. Stigma's negative impact on the health and quality of life of people living with HIV and key populations is also well documented [3,9–14].

Stigma is a social process in which an individual or group is linked to a negative stereotype or misconception, often resulting in adverse experiences, loss of social status, and limited opportunities [15,16]. Stigma may occur at the individual, interpersonal, community, and structural levels and can be experienced as anticipated, perceived, internalized, or enacted

[17]. Anticipated stigmas refer to the expectation of bias perpetrated by others [17–20]. Perceived stigmas refer to felt stigma and the perception of bias as understood by a person living with a stigmatized identity [21]. Internalized, or self-stigma, is the adoption of negative feelings or devaluing of oneself on account of a stigmatized identity [17,22]. Enacted, or experienced stigma, is the perpetration of mistreatment or discriminatory acts by someone on the basis of a stigmatized identity [23]. Discrimination, part of the social process of stigma [15], includes any distinction, exclusion, or restriction made to human rights and fundamental freedoms, in the political, economic, social, cultural, civil, or any other field [24]. Discrimination can be institutionalized through existing laws, policies, or practices that negatively impact people living with HIV or key populations. Although the conceptualization of discrimination in relation to stigma varies, here, we consider enacted stigma to be inclusive of discrimination [17]. Drivers of HIV and key population stigmas can include individual-level factors such as lack of awareness or education (i.e., misinformation about HIV risk and transmission); however, societal level policies, cultural norms, and religious values can also act as drivers or facilitators of stigma [23,25–27].

In recognition of the central role stigma has in impeding HIV epidemic transition, UNAIDS has established a vision to achieve 3 zeros by 2030: zero new HIV infections, zero AIDS-related deaths, and zero discrimination [2,9]. To support this vision, UNAIDS has also established the 10–10–10 goals that focus on removing societal and legal barriers to HIV services, including reductions in punitive laws and policies, experiences of stigma and discrimination, and experiences of gender inequality and violence by 2025 [28]. Stigma related to HIV and key population statuses have also been recognized as a key factor impeding progress toward the 2030 Agenda for Sustainable Development Goal (SDG 3.3) of achieving the end of the HIV pandemic [24,29]. Despite this consensus on the importance of addressing stigma, measurement of stigma experienced by people living with HIV and key populations continues to be a challenge for public health practitioners and policymakers [18,30]. Stigma measurement has unique challenges given that it is a latent construct, a social process that acts across multiple levels, and a phenomenon that has often been thought of as too complex to measure [31–33]. However, progress has been made over the last decade in developing standardized measures to quantify both HIV stigma and key population stigma [18,34–38]. Specifically, advancements have been made through the Global AIDS Monitoring indicators and WHO Strategic Information guidelines [39,40]. Despite the existence of many validated measures of different forms of stigma related to HIV and key population membership, there is currently no established methodology for bringing these together into more concise but validated summary measures at the country level. A summary measure to quantify progress toward achieving zero discrimination may allow for easy comparisons within and across countries and possibly improved accountability for progress.

The ability to characterize stigma across countries was recommended as a strategy to achieve the end of the HIV pandemic during the 2017 UNAIDS Science Panel meeting on ending the AIDS epidemic by 2030 [41]. In response, the UNAIDS Monitoring Technical Advisory Group (MTAG) created a task team in 2018 comprised of technical experts on HIV and key population stigma from civil society and academia from around the globe. This task team's mission was to provide a set of consensus recommendations to UNAIDS on the establishment of a country-level summary measure of stigma faced by people living with HIV and key populations at high risk of HIV and the legal and policy environment for the protection of these individuals' fundamental rights, including their ability to access health and HIV services.

The creation of country-level indices that characterize the degree of stigma existing within a country may facilitate understanding and interpreting the overall burden of stigma and its effects. For the purposes of this work, we consider an indicator to be a single item measuring

an aspect of stigma, and an index is a composite of indicators to generate a summary or average value representing a broader stigma construct. Summary index measures may also be used by global- and national-level policymakers, advocates, and program implementers engaged in the AIDS response to track progress on eliminating HIV and key population stigma, potentially increasing the likelihood that this goal can and will be achieved. The question remains, however, whether sufficient data exist to inform country-level indices that characterize levels of stigma for people living with HIV and key populations. This study aims to identify and describe the availability and quality of stigma data across countries through a global data mapping exercise.

## Methods

The methods used in this exercise, including preparatory activities for data mapping and the mapping of available data for country-level summary measures of HIV and key population stigma, are outlined in Fig 1. The protocol for this exercise is included as S1 Protocol.

### Task team and consultation process in preparation of data mapping

UNAIDS Strategic Information Department led an initial drafting of possible domains, subdomains, and indicators to be considered for inclusion in summary measures of stigma affecting people living with HIV and 4 key populations: men who have sex with men, sex workers, people who use drugs, and transgender people. In drafting indicators, established definitions, such as those created as a part of the Global AIDS Monitoring process [39], were used when available. The long-term goal of this process is to generate summary measures using existing indicators as developing new indicators would take significant time to finalized and generate data. Therefore, this initial consultation process only considered existing indicators that had been defined and validated or existing validated data collection tools.

The designated task team provided feedback on UNAIDS' initial selection and helped guide a 3-week e-consultation that took place in August to September 2019. The consultation

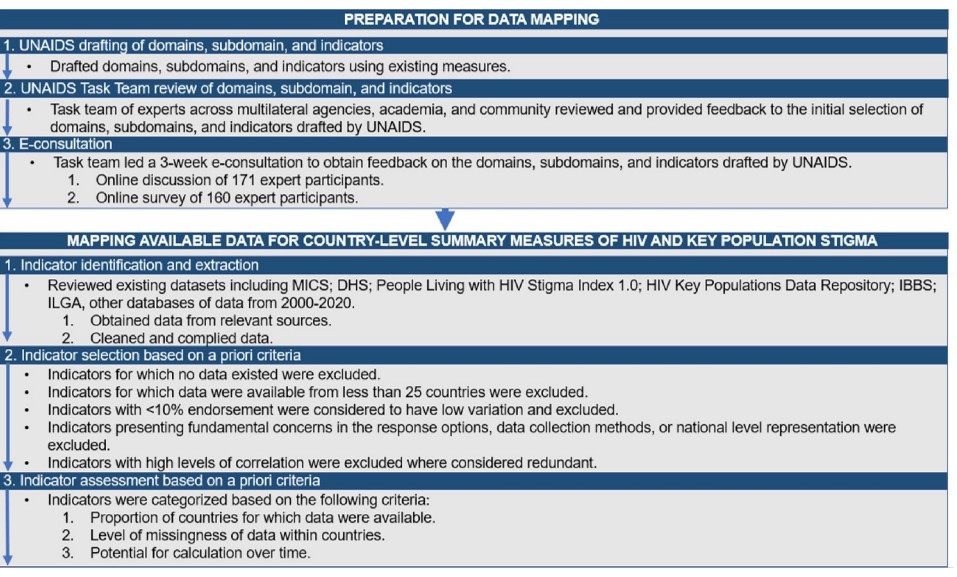

**Fig 1. Flowchart of methodological approach for data preparation and data mapping.** DHS, Demographic and Health Surveys; IBBS, Integrated Biological and Behavioral Surveys; ILGA, International Lesbian, Gay, Bisexual, Trans and Intersex Association; MICS, Multiple Indicator Cluster Surveys.

information was shared through UNAIDS mailing lists, social media, and partners. Interested participants were invited to register on the consultation page. Overall, 804 people registered, of whom 171 participated in online discussions and 160 in a short survey on their background. Individuals participated from different global regions, including individuals from HIV or key population communities, nongovernmental organizations, civil society or community-based or faith-based organizations, academia, government ministries, and the private sector or professional organizations. Descriptive characteristics of participants in e-consultation online survey are outlined in S1 Table. During the e-consultation, participants were invited to engage in online discussions on key questions structured around the draft domains, subdomains, and indicators, as well as how to bring these together.

## Mapping available data for country-level summary measures of HIV and key population stigma

This data mapping exercise was an exploratory process. The criteria used to identify, include, and assess indicators were established a priori. However, final decisions were informed by the data obtained as part of a larger plan to understand stigma metrics.

**Indicator identification and extraction.** As described, UNAIDS considered existing tracking systems and data sources in the initial identification and selection of indicators. However, a need to identify the level of data availability and quality for each proposed indicator remained in order to determine the feasibility of inclusion in measure development. In collaboration with UNAIDS, the research team conducted a data mapping exercise to determine the number of countries with sufficient and appropriate data that could be used to calculate each indicator over the period of 2000 to 2020. As a first step, the research team reviewed relevant datasets including the UNAIDS National Commitments and Policies Instrument (NCPI) database [42]; Multiple Indicator Cluster Surveys (MICS) [43]; Demographic and Health Surveys (DHS) [44]; People Living with HIV Stigma Index 1.0 surveys [45]; HIV Key Population Data Repository; a selection of Integrated Biological and Behavioral Surveys (IBBS) available to the research team [4,37]; and network databases. This entailed downloading relevant NCPI data from the UNAIDS databases and MICS data from the UNICEF webpage. DHS surveys were pulled from STATcompiler, and individual country DHS reports were reviewed as relevant. The research team worked directly with the Global Network of People Living with HIV (GNP +) to access and combine individual files of 34 country datasets of the People Living with HIV Stigma Index 1.0. GNP+ is currently leading a People Living with HIV Stigma Index 2.0, and, therefore, the survey for this study was also reviewed as were plans for its implementation. The IBBS studies considered in this exercise were implemented by the Key Populations Program at Johns Hopkins School of Public Health and therefore the research team had access to these data. IBBS data were obtained from individual studies conducted among men who have sex with men or sex workers in sub-Saharan Africa where possible based on the team's preexisting access as well as review of files that summarized existing literature provided by UNAIDS. The HIV Key Populations Repository, which is a database developed from a comprehensive review of all available data for key populations, including burden and risk of HIV, prevalence, incidence, prevention indicators and treatment cascades, population size estimates, experienced violence, and engagement with healthcare systems was reviewed [46]. Additionally, databases from networks in including the International Lesbian, Gay, Bisexual, Trans and Intersex Association (ILGA), HIV Justice Network, Advancing HIV Justice were reviewed [47]. Other databases were reviewed to assess data availability for specific indicators collected outside of health-focused surveys as necessary. Data extraction was conducted between August and November 2020.

**Indicator selection.** After data were extracted from these sources, further cleaning was conducted. This included reverse coding to ensure consistent direction of endorsement where necessary and the identification of outliers. With the extracted data, data aggregated to the country level was combined to create a country-level database. Descriptive statistics were calculated for all indicators with existing data, including, number, proportion, or percentage, as well as mean and SD when appropriate. Importantly, not all estimates in this dataset are nationally representative, given that data points are subject to individual study design and recruitment strategies.

Indicators were then selected based on a set of a priori criteria, outlined below, for their potential for creating a summary measure, also known as an index. Indicators for which no existing data source was identified for any country were removed from consideration. Indicators for which data were identified were further assessed based on (1) number of countries for which data were available for the indicator; (2) variation present across countries in the indicator; and (3) data quality or accuracy. Indicators for which data were available from less than 25 countries were removed. We examined variable distributions to both determine if any data transformations were necessary but also to assess whether sufficient variation in values existed to inform a composite index. Variation in the indicator was determined based on the proportion of affirmed responses. Indicators with less than 10% endorsement were considered to have low variation and therefore limited utility in the creation of an index. Indicators with low variation were removed from further consideration for use at present in empirical index creation processes. Data quality and accuracy of indicators were assessed to determine if there were fundamental concerns in the response options, data collection methods, or national-level representation.

Last, a correlation matrix of all indicators was generated to assess potential redundancy. While we expected correlation between indicators, a very high degree of correlation between 2 indicators was taken to be suggestive of redundancy that should be eliminated to create the most parsimonious and informative measure [48]. A pairwise correlation matrix was generated between all stigma indicators related to HIV or key population status, and a Pearson correlation coefficient (rho) of greater than 0.6 was used as indication of a potential redundancy, i.e., where limited additional information would be added to an index given the inclusion of other indicators. Indicators determined to be largely redundant based on high intercorrelation with other indicators were considered for removal or to be combined into a single measure.

**Indicator assessment.** Indicators that met the criteria for inclusion were further assessed for potential use in the creation of indices at present based on (1) proportion of countries for which data were available; (2) level of missingness of data within countries; and (3) potential for calculation over time. The proportion of countries for which data were available for the indicator was determined to be "sufficient" if it was available in 50% or more of the 193 UN member states. Indicators that were available in less than 50% of countries were designated as "limited." The cutoff of 50% was meant to demonstrate data availability for the majority of UN member states (countries). Within countries, those which had data available for greater than 40% of the identified indicators, these countries were labeled as having "sufficient" available data; where countries had data available for less than 40% of the indicators, they were labeled as having "limited" available data. An indicator was also assessed for potential use in assessing changes over time. Specifically, if an indicator had more than 1 time point of available data and had not fundamentally changed in its definition over time, it was considered as "potentially" useful for tracking changes over time. If there were changes in the indicator over time, it was determined to have "limited" utility, and, if data for the indicator was available for only 1 time point with no planned future data collection, it was categorized as "cannot" track changes over time.

**Table 1. Summary of stigma indicator mapping identification, extraction, selection, and assessment.**

| | HIV related | Key population related | | | |
|---|---|---|---|---|---|
| | HIV | Men who have sex with men | Sex work | Drug use | Transgender |
| **Initial indicators based on UNAIDS, e-consultation, and research team review** | **44** | **14** | **12** | **10** | **17** |
| *Removed due to lack of existing data* | 9 | 7 | 5 | 7 | 11 |
| **Available data sources** | **35** | **7** | **7** | **3** | **6** |
| *Removed due to limited number of countries for which data were available for the indicator* | 3 | 4 | 4 | 1 | 1 |
| *Removed due to variation present across countries in the indicator* | 3 | 0 | 0 | 0 | 0 |
| *Removed due to systematic concerns over data quality or accuracy* | 4 | 0 | 0 | 0 | 0 |
| **Recommended indicators for weight calculation or summary measures** | **25** | **3** | **3** | **2** | **5** |
| *Collapsed measures due to data reporting or redundancy* | 2 | 2 | 2 | 0 | 2 |
| **Final indicators** | **24** | **2** | **2** | **2** | **4** |

## Ethical approval

The Johns Hopkins School of Public Health Institutional Review Board (JHSPH-IRB) reviewed this data mapping exercise and secondary data analysis protocol and determined it to be nonhuman subjects research as only deidentified data were being used.

## Results

The summary results of indicator identification, extraction, selection, and assessment for HIV stigma indicators and the key population stigma indicators are presented in Table 1.

## E-consultation

The e-consultation process resulted in a revised list of indicators that covered 6 domains: (1) structural stigmas; (2) social norms and attitudes reported among the general community; (3) anticipated stigma; (4) experienced stigma; (5) internalized stigma, and (6) experiences of violence as reported by people living with HIV or key populations. The full list if proposed indicators are described in S2–S6 Tables. The structural stigma domain was originally proposed by UNAIDS as a laws and policies domain. As a result of data identified in this mapping process, we expanded this domain to include access to justice and renamed the domain accordingly. Additionally, anticipated and experienced stigmas were originally included in a single domain, however, were separated into 2 separate domains to represent these distinct types of stigmas. Five summary stigma measures were proposed that would be able to describe (1) stigma affecting people living with HIV; (2) sexual minority stigma (including sexual behavior or orientation) related to men who have sex with men; (3) sex work stigma; (4) drug use stigma; and (5) gender identity stigma related to transgender persons. Across the 5 summary measure categories, the subdomains and the exact indicators included varied. In total, 44 indicators were proposed for HIV stigma, and for key populations, 14 were proposed for men who have sex with men; 12 for sex work; 10 for drug use; and 17 for transgender persons.

## Data availability and quality

We were able to identify a data source and existing data for 39 of the 44 HIV stigma indicators selected in the e-consultation process (Table 1). Data for 4 of these indicators were available from the People Living with HIV Stigma Index 1.0 survey; however, the corresponding questions were slated for exclusion from the 2.0 version of the survey, and, thus, these indicators were

excluded from further consideration for use in the summary measure. This left a total of 35 indicators for HIV stigma in consideration, of which another 10 were removed: 3 due to the limited number of countries for which data were available; 3 due to limited variability; and 4 due to inconsistency in data collection and/or fundamental concerns about data quality or access.

In terms of key populations, data sources with existing available data were only found for 7 of 14 indicators for sexual minority stigma related to men who have sex with men, 7 of 12 indicators for sex work stigma, 3 of 10 indicators for drug use stigma, and 6 of 17 indicators for gender identity stigma related transgender persons (Table 1). We recommended changes to the time frame of 2 indicators to be consistent with available potential data sources. Outside of HIV stigma, no data were identified for indicators within the domains of social norms and attitudes, violence, and internalized stigma. Additionally, no indicators within the domain of stigma and discrimination had sufficient data across countries or planned data collection for measurement over time for key populations. Therefore, these domains could not be considered for stigmas related to key populations after completing data mapping at this time. Across key populations, we were only able to retain 3 of 7 men who have sex with men indicators, 3 of 7 sex work stigma indicators, 2 of 3 drug use stigma indicators, and 5 of 6 gender identity stigma indicators due to limited data availability across countries.

## Interrelationship of indicators

Several sets of indicators were combined due to how the data were reported. Among indicators for sexual minority stigma related to men who have sex with men, the indicators "Existence of constitutional protections of discrimination or other nondiscrimination provisions related to sexual orientation" and "Existence of laws or other provisions that prohibit discrimination in employment based on sexual orientation" were assessed via items that shared a "No" response option and are therefore collapsed into a single measure. Among indicators for sex work stigma, "Existence of constitutional protections of discrimination based on occupation or other nondiscrimination provisions specifying sex work" and "Existence of laws or policies recognizing sex work as work" are collapsed into a single measure as these items were also collected with a shared "No" response option. Similarly, 2 of the 5 indicators for gender identity stigma with data available from more than just a few countries had to be combined due to how the data were reported through the NCPI (i.e., the questionnaire was structured so that indication of presence or absence of laws or policies assessed in these indicators shared a "None of these policies" category). These indicators were collapsed into a single measure.

For HIV stigma, the indicator "Percentage of people living with HIV who have lost a source of income or job because of their HIV status in the past 12 months" had a Pearson correlation coefficient of 0.74 with "Percentage of people living with HIV who have been refused employment or a work opportunity because of their HIV status in the past 12 months," suggesting that inclusion of both of these indicators may not be necessary due to potential redundancy. We recommend retaining the former and excluding the latter given the former has data available for one additional country. For future assessment, PLHIV Stigma Index 2.0 index combines these 2 indicators which resolves this issue going forward [45]. These same indicators were correlated with "Percentage of people living with HIV who experienced social exclusion in the last 12 months due to their HIV status" (rho = 0.82 and 0.84, respectively). "Feeling shame and guilt" and "Avoiding healthcare out of fear of discrimination" were also correlated at rho = 0.63. Although these indicators are correlated, they are not from the same domain, and, therefore, it is not recommended that they are collapsed or removed at this stage.

Among key population indicators, the sexual minority and gender identity "Discrimination and employment protection policies" indicators were correlated at a rho = 0.68; however,

given these are different key populations (men who have sex with men; transgender persons), these indicators were retained at this stage. Indicators with a correlation of rho <0.60 were not considered for combination. Although some indicators showed correlation, no indicators were removed for redundancy at this stage.

### Final potential indicators

This mapping exercise identified 24 potential HIV stigma indicators and 10 key population indicators with potential for use in characterizing stigma and creating valid stigma summary measures. The key population indicators include 2 for sexual orientation/behavior stigma related to men who have sex with men; 2 for sex work-related stigma; 2 for drug use stigma; and 4 for gender identity stigma related to transgender people. These are described in detail in Tables 2–6.

### Availability across countries

For the 24 remaining indicators of HIV stigma, 11 (45.8%) were identified as having data available in more than half of the 193 UN member states (Table 2). Among the 10 key population indicators under consideration (Tables 3–6), data were available for all in more than 50% of UN member states.

### Geographic representation

We found 61 of 193 UN member states with any available data (32%) to have data for a sufficient number of HIV stigma indicators (missing data for less than 40% of indicators). Considering all key population stigma indicators together, 119 of 193 countries (62%) had data on a sufficient number of indicators. Countries that meet the criteria of having sufficient data across the 24 HIV indicators are displayed in Fig 2 and for the 10 key population indicators in Fig 3.

### Forms of stigma

Summary of results from existing data for the HIV stigma indicators are presented in Table 2. Overall, 44.3% (SD = 20) of women and men 15 to 49 years old report discriminatory attitudes as measured by a composite of 2 questions. The degree of criminalization of HIV and key population associated behaviors varied: 61.7% ($N$ = 92/191) of countries reported existence of laws criminalizing the transmission of, nondisclosure of, or exposure to HIV transmission (Table 2); 35.2% (68/193) of countries reported the existence of laws criminalizing consensual same-sex sexual acts (Table 3); 85.8% (121/141) of countries reported the existence of laws criminalizing sex work or with any punitive measures related to sex work (Table 4); 81.7% (98/120) of countries report the existence of laws criminalizing drug use and/or possession for personal use (Table 5); and 18.5% (25/135) of countries reported existence of laws criminalizing transgender people and/or cross-dressing (Table 6).

### Tracking change over time

Of the 24 potential HIV stigma indicators, 19 were found to have potential for use in tracking change in stigma over time, which includes 11 HIV stigma indicators which had data available in more than half of UN member states. All the final 10 stigma indicators for key population stigma, including sexual minority stigma, sex work stigma, drug use stigma, and gender identity stigma related to transgender persons have the potential to be used to track change over time.

**Table 2. Final indicators for HIV stigma.**

| Domain | Subdomain | # | Indicator | Countries with available data | Possible to measure change over time | Descriptive statistics Number (%) or mean (SD) and range | Data source |
|---|---|---|---|---|---|---|---|
| Social norms and attitudes | Discriminatory attitudes toward people living with HIV | 1 | Percentage of women and men 15 to 49 years old who report discriminatory attitudes (composite of 2 questions) | 32 for both questions; 55 for one | Potentially | Mean (SD) = 44.4% (20.4%); Range: 5.7% to 81.4% | DHS, MICS |
| | Acceptability of partner violence | 2 | Percentage of all women and men who agree that a husband is justified in hitting or beating his wife for specific reasons | 71 (women); 62 (men) | Limited | Mean (SD) = 29.3% (15.6%); Range: 3.9% to 72.4% | DHS |
| Structural stigma | Selective and arbitrary arrest and prosecution | 3 | Existence of laws criminalizing the transmission of nondisclosure of, or exposure to HIV transmission | 149 | Limited | N (%): No = 30 (20.1%); No, but prosecutions exist based on general criminal laws = 27 (18.1%); Yes = 92 (61.7%) | UNAIDS NCPI; Advancing HIV Justice |
| | | 4 | Number of prosecutions for HIV transmission | 191 | Potentially | N (%): No cases reported = 145 (75.9%); 1 to 2 reported cases = 30 (15.7%); Fewer than 1/10,000 cases reported = 4 (2.1%); Between 1/1,000 and 1/10,000 cases reported = 9 (4.7%); Greater than or equal to 1/1,000 cases reported = 3 (1.6%) | HIV Justice Network |
| | Restrictions on entry, stay, or residence | 5 | Existence of laws restricting the entry, stay, and residence of people living with HIV | 191 | Cannot | N (%): No restriction = 147 (77.0%); Require testing or disclosure = 16 (8.4%); Prohibit stays = 10 (5.2%); Deport = 18 (9.4%) | UNAIDS NCPI |
| | Mandatory testing | 6 | Existence of laws, regulations, or policies specifying HIV testing is mandatory before marriage, to obtain a work or residence permit and/or for certain groups | 144 | Potentially | N (%): No mandatory testing laws = 47 (32.6%); one mandatory testing law = 70 (48.6%); 2 mandatory testing laws = 20 (13.9%); 3 mandatory testing laws = 7 (4.9%) | UNAIDS NCPI |
| | Consent to access sexual and reproductive health and HIV services | 7 | Existence of laws requiring parental/guardian consent for adolescents to access HIV testing and receive the results | 141 | Potentially | N (%): No law = 36 (25.5%); Required for age <12 = 1 (0.7%); Required for age <14 = 29 (20.6%); Required for age <16 = 28 (19.9%); Required for age <18 = 47 (33.3%) | UNAIDS NCPI |
| | | 8 | Existence of laws requiring parental/guardian consent for adolescents to access HIV treatment | 137 | Potentially | N (%): No law = 52 (38.0%); Required for age <14 = 17 (12.4%); Required for age <16 = 21 (15.3%); Required for age <18 = 47 (34.3%) | UNAIDS NCPI |
| | | 9 | Existence of laws requiring parental/guardian consent for adolescents to access contraceptives, including condoms | 90 | Potentially | N (%): No law = 46 (51.1%); Required for age <12 = 3 (3.3%); Required for age <14 = 9 (10.0%); Required for age <16 = 6 (6.7%); Required for age <18 = 26 (28.9%) | UNAIDS NCPI |
| | | 10 | Existence of laws requiring spousal consent for married women to access any sexual or reproductive health service | 142 | Potentially | N (%) with a law: 9 (6.3%) | UNAIDS NCPI |
| | Nondiscrimination | 11 | Existence of laws or policies requiring healthcare settings to provide timely and quality healthcare regardless of any grounds | 131** | Potentially | N (%): No policy exists = 4 (3.1%); Yes, policy exists but is not consistently implemented = 41 (31.3%); Yes, policy exists and is consistently implemented = 86 (65.7%) | UNAIDS NCPI |
| | | 12 | Existence of laws protecting against discrimination on the basis of HIV status | 88 | Potentially | N (%): No law = 17 (19.3%); Yes, HIV protected under another status = 30 (34.1%); Yes, HIV specified as protected attribute = 41 (46.6%) | UNAIDS NCPI |
| | | 13 | Existence of government mechanisms to record and address individual complaints cases of HIV-related discrimination (based on perceived HIV status and/or belonging to any key population) | 129*; 126** | Potentially | N (%) with law: - National authority report = 87 (69.1%); - Civil society report = 86 (66.7%) | UNAIDS NCPI |
| Violence (physical, sexual, emotional/ psychological, and economic) | Controlling partner behaviors | 14 | Percentage of ever-married women whose husbands/partners demonstrated types of controlling behaviors | 54 | Potentially | Mean (SD): Percentage of respondents reporting 3 or more controlling behaviors: 22.5% (10.8%), Range: 5.4% to 51.8% | DHS |
| | Recent experience of violence | 15 | Percentage of women age 15 to 49 who have experienced physical and/or sexual violence by an intimate partner in the past 12 months | 55 | Potentially | Mean (SD): Percentage of female respondents who have experienced physical and/or sexual violence by an intimate partner: 19.3% (10.1%), Range: 3.5% to 47.6% | DHS |
| | National policy environment | 16 | Existence of a national plan or strategy to address gender-based violence and violence against women that includes HIV | 127 | Potentially | N (%) with a law: 105 (82.7%) | UNAIDS NCPI |
| | | 17 | Existence of specific legal provisions prohibiting violence against people based on their HIV status or belonging to a key population | 122*; 124** | Potentially | N (%) with provisions: - National authority report = 55 (44.4%); - Civil society report = 58 (47.5%) | UNAIDS NCPI |

(*Continued*)

**Table 2.** (Continued)

| Domain | Subdomain | # | Indicator | Countries with available data | Possible to measure change over time | Descriptive statistics Number (%) or mean (SD) and range | Data source |
|---|---|---|---|---|---|---|---|
| | Experience of violence in healthcare settings | 18 | Percentage of people living with HIV who were forced, pressured, or paid/incentivized to get sterilized and/or advised to terminate a pregnancy | 35 | Potentially | Mean (SD) = 6.2% (4.3%), Range: 0% to 18.5% | PLHIV Stigma Index |
| Anticipated stigma | Discrimination anticipated in healthcare settings | 19 | Percentage of people living with HIV who avoided seeking healthcare in the past 12 months due to fear of stigma and discrimination | 34 | Potentially | Mean (SD) = 16.0% (9.6%), Range: 2.9% to 41.7% | PLHIV Stigma Index |
| Experienced stigma | Social exclusion | 20 | Percentage of people living with HIV who experienced social exclusion in the last 12 months due to their HIV status | 33 | Potentially | Mean (SD) = 16.2% (10.4%), Range: 4.2% to 45.0% | PLHIV Stigma Index |
| | Discrimination experienced in healthcare settings | 21 | Percentage of people living with HIV who report experiences of HIV-related discrimination in healthcare settings | 36 | Limited | Mean (SD) = 19.5% (12.9%), Range: 3.7% to 53.1% | PLHIV Stigma Index |
| | Discrimination experienced in employment | 22 | Percentage of people living with HIV who have lost a source of income or job because of their HIV status in the past 12 months | 36 | Potentially | Mean (SD) = 10.2% (8.0%), Range: 1.5% to 31.8% | PLHIV Stigma Index |
| | | | Percentage of people living with HIV who have been refused employment or a work opportunity because of their HIV status in the past 12 months*** | 35 | Potentially | Mean (SD) = 10.9% (6.8%), Range: 2.9% to 30.9% | PLHIV Stigma Index |
| Internalized stigma | Social isolation | 23 | Percentage of people living with HIV that report self-isolating from others | 34 | Potentially | Mean (SD) = 26.7% (13.4%), Range: 5.8% to 53.8% | PLHIV Stigma Index |
| | Negative self-beliefs or feelings | 24 | Percentage of people living with HIV that report shame or guilt | 35 | Limited | Mean (SD) = 54.8% (15.4%); Range: 26.3% to 82.0% | PLHIV Stigma Index |

*Civil society report.

**National authority report.

***Future data collection for PLHIV Stigma Index 2.0 combines "Percentage of people living with HIV who have lost a source of income or job because of their HIV status in the past 12 months" and "Percentage of people living with HIV who have been refused employment or a work opportunity because of their HIV status in the past 12 months," and this combined indicator is recommended for future use.

DHS, Demographic and Health Surveys; MICS, Multiple Indicator Cluster Surveys; NCPI, National Commitments and Policies Instrument; PLHIV, people living with HIV.

## Discussion

In 2022, data exist for comprehensively characterizing the nature and level of different forms of stigma affecting people living with HIV and key populations. However, the data mapping exercise described here highlighted a lack of stigma indicators that have ongoing, consistent data collection for assessing change over time, and thus the ability to track progress in mitigating stigma at the country level. In particular, data for indicators outside of the structural stigma domain are much less consistently and rigorously collected and made accessible, particularly for stigma experienced by key populations. There are a few indicators at present that have existing data and are incorporated within specific frameworks for routine data collection mechanisms. Given geographic representation, repeated implementation, and quality, these indicators provide some ability for beginning to explore if and how complex experiences of stigma may be able to be summarized to elucidate the state of stigma within a country and assess change going forward. Despite the current limitations in data availability and quality, participatory approaches involving experts from communities, civil society, academia, and public health practices can be used to help inform the creation of summary measures (i.e., via use of participatory weighting) and to inform data collection priorities moving forward (i.e., via participatory ranking of indicator importance).

Unfortunately, several domains, subdomains, and indicators of stigma were not identified as having sufficient data to use in a summary measure at present despite their importance. No publicly available data sources for appropriately assessing indicators within the domains of

**Table 3. Final indicators for sexual behavior/orientation stigma related to men who have sex with men.**

| Subdomain | # | Indicator | Countries with available data | Possible to measure change over time | Descriptive statistics Number (%) or mean (SD) and range | Data source |
|---|---|---|---|---|---|---|
| Criminalization of same-sex sexual acts | 1 | Existence of laws criminalizing consensual same-sex sexual acts | 193 | Potentially | N (%) with criminalization: 68 (35.2%) | UNAIDS NCPI; ILGA |
| Nondiscrimination laws | 2 | Existence of constitutional protections of discrimination or other nondiscrimination provisions related to sexual orientation | 87*; 84** | Potentially | N (%) with prohibition:<br>- National authority report = 45 (53.6%);<br>- Civil society report = 58 (66.7%) | UNAIDS NCPI; ILGA |
| | | Existence of laws or other provisions that prohibit discrimination in employment based on sexual orientation | 76*; 80** | Potentially | N (%) with prohibition:<br>- National authority report = 43 (53.8%);<br>- Civil society report = 44 (57.9%) | UNAIDS NCPI; ILGA |

*Civil society report.

**National authority report.

ILGA, International Lesbian, Gay, Bisexual, Trans and Intersex Association; NCPI, National Commitments and Policies Instrument.

social norms and attitudes, or internalized stigma for key populations were identified, and, therefore, these domains had to be removed from key population stigma summary measures at present. Given that social norms and attitudes have been well documented to play a large role in the experiences of stigma among key populations and to be associated with health outcomes, data collection for indicators within this domain should be prioritized in order to include this domain when characterizing stigma at a country level. Indicators within the violence, anticipated stigma, experienced stigma, and stigma and discrimination domains were not retained

**Table 4. Final indicators for sex work stigma.**

| Domain | Subdomain | # | Indicator | Countries with available data | Possible to measure change over time | Descriptive statistics Number (%) or mean (SD) and range | Data source |
|---|---|---|---|---|---|---|---|
| Structural stigma | Criminalization of sex work | 1 | Existence of any criminalization of sex work | 141 | Potentially | N (%)<br>With no criminalization = 20 (14.2%);<br>Criminalization of buying and/or selling sex = 77 (54.6%);<br>Partial criminalization = 15 (10.6%); other ancillary or punitive measures = 29 (20.6%) | UNAIDS NCPI; ILGA |
| | Nondiscrimination laws | 2 | Existence of constitutional protections of discrimination based on occupation or other nondiscrimination provisions specifying sex work | 119*; 123** | Potentially | N (%) with prohibition:<br>- National authority report = 20 (16.3%);<br>- Civil society report = 28 (23.5%) | UNAIDS NCPI |
| | | | Existence of laws or policies recognizing sex work as work | 107*; 109** | Potentially | N (%) with recognition:<br>- National authority report = 4 (3.7%);<br>- Civil society report = 9 (8.4%) | UNAIDS NCPI |

*Civil society report.

**National authority report.

ILGA, International Lesbian, Gay, Bisexual, Trans and Intersex Association; NCPI, National Commitments and Policies Instrument.

**Table 5. Final indicators for drug use stigma.**

| Domain | Subdomain | # | Indicator | Countries with available data | Possible to measure change over time | Descriptive statistics Number (%) or mean (SD) and range | Data source |
|---|---|---|---|---|---|---|---|
| Structural stigma | Criminalization of drug use and/or possession | 1 | Existence of laws criminalizing drug use and/or possession for personal use | 120 | Potentially | N (%) criminalized = 98 (81.7%) | UNAIDS NCPI |
| | Nondiscrimination laws | 2 | Existence of any specific anti-discrimination laws or other protective provisions that apply to people who use drugs | 125*; 128** | Potentially | N (%): - National authority report = 17 (13.3%); - Civil society report = 15 (12%) | UNAIDS NCPI |

*Civil society report.

**National authority report.

NCPI, National Commitments and Policies Instrument.

for consideration as a summary measure for key populations due to limited availability and quality. Therefore, the potential indicators for key population–related stigma only included those from the structural stigma domain. This highlights the importance of expanding data collection for these domains to improve both quality and availability of data that can be used to more comprehensively characterize stigma as it exists in different forms for key populations. Guidance on biobehavioral surveys have been recently developed and can be used by countries to improve consistent data collection of validated measures in the future [49].

Often, only data aggregated at the sample or even country level were able to be located for indicators, which presents a challenge in accounting for differences in sampling strategies when comparing across countries. For instance, individual-level indicators assessing attitudes

**Table 6. Final indicators for gender identity stigma related to transgender persons.**

| Domain | Subdomain | # | Indicator | Countries with available data | Possible to measure change over time | Descriptive statistics Number (%) or mean (SD) and range | Data Source |
|---|---|---|---|---|---|---|---|
| Structural stigma | Criminalization or prosecution | 1 | Existence of laws criminalizing transgender people and/or cross-dressing | 135 | Potentially | N (%) with a law: 25 (18.5%) | UNAIDS NCPI |
| | Nondiscrimination laws | 2 | Existence of constitutional protections of discrimination or other nondiscrimination provisions related to gender diversity | 126*; 125** | Potentially | N (%): with a law: - National authority report = 51 (40.8%); - Civil society report = 58 (46.0%) | UNAIDS NCPI |
| | | | Existence of laws or other provisions that prohibit discrimination in employment based on gender diversity | 114*; 103** | | N (%): with a law - National authority report = 40 (38.8%); - Civil society report = 47 (41.2%) | UNAIDS NCPI |
| | | 3 | Existence of legislation allowing gender marker change | 134 | Potentially | N (%): not possible = 48 (35.8%); possible nominally or with restriction or lack of clarity = 51 (38.1%); possible = 35 (26.1%) | ILGA |
| | | 4 | Existence of legislation allowing name change | 130 | Potentially | N (%): not possible = 24 (18.5%); possible nominally or with restriction or lack of clarity = 16 (12.3%); possible = 90 (69.2%) | ILGA |

*Civil society report.

**National authority report.

ILGA, International Lesbian, Gay, Bisexual, Trans and Intersex Association; NCPI, National Commitments and Policies Instrument.

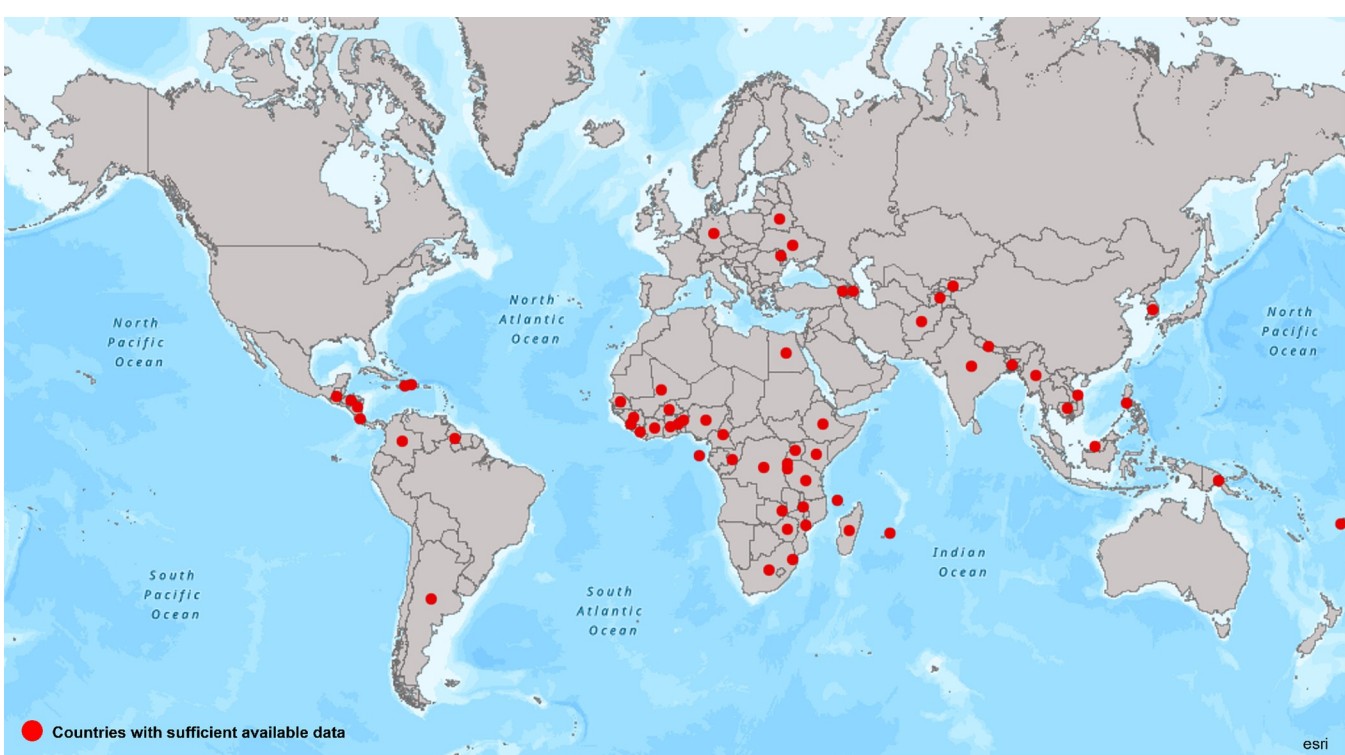

**Fig 2. Countries with sufficient available data on HIV stigma indicators.** Base layer of the map used are from ArcGIS, ESRI: https://arcg.is/8DHLK. *Data were available for more than 40% of indicators.

or experiences of discrimination are often reported at country level, although data may not have been drawn from a nationally representative sample. Statistical methods can be used to better account for sampling approaches to address these concerns of representativeness; however, this is not possible without access to individual-level data, and neither is it possible to recombine or disaggregate samples (such as by age or gender). While there may be consistency in how individual-level surveys are collected, without systems for sharing deidentified data from such surveys, it is not possible to use these data to inform broader comparative analyses aimed at assessing country progress. This limits availability for indicator calculation. The creation of a broader data repository for individual-level deidentified data provides an opportunity to address both of these concerns and keep data for indicators easily accessible for monitoring and analysis. Countries could then better understand the data available and use weighting strategies appropriately to characterize stigma even in the context of limited data. When stigma is measured in surveys, especially when stigmas affecting key populations are assessed, validated measures or scales are often not used, and there is great variety in the measures used across studies [18]. Therefore, an essential step in improving usability of data shared in a repository is increasing the consistency and standardization of stigma measures across countries to facilitate improved comparability of secular trends and health outcomes of stigma.

Stigma mitigation can emerge from both interventions within and outside of the health sector [23]. Measuring secular trends with standardized measures across and within settings can provide insight into the potential impact of interventions aimed at addressing human rights barriers and complementary investments aiming to address stigma [30]. A quantitative measure of stigma, which links directly to diminished uptake of HIV prevention or treatment services, offers an additional indicator against which to measure progress in the AIDS response.

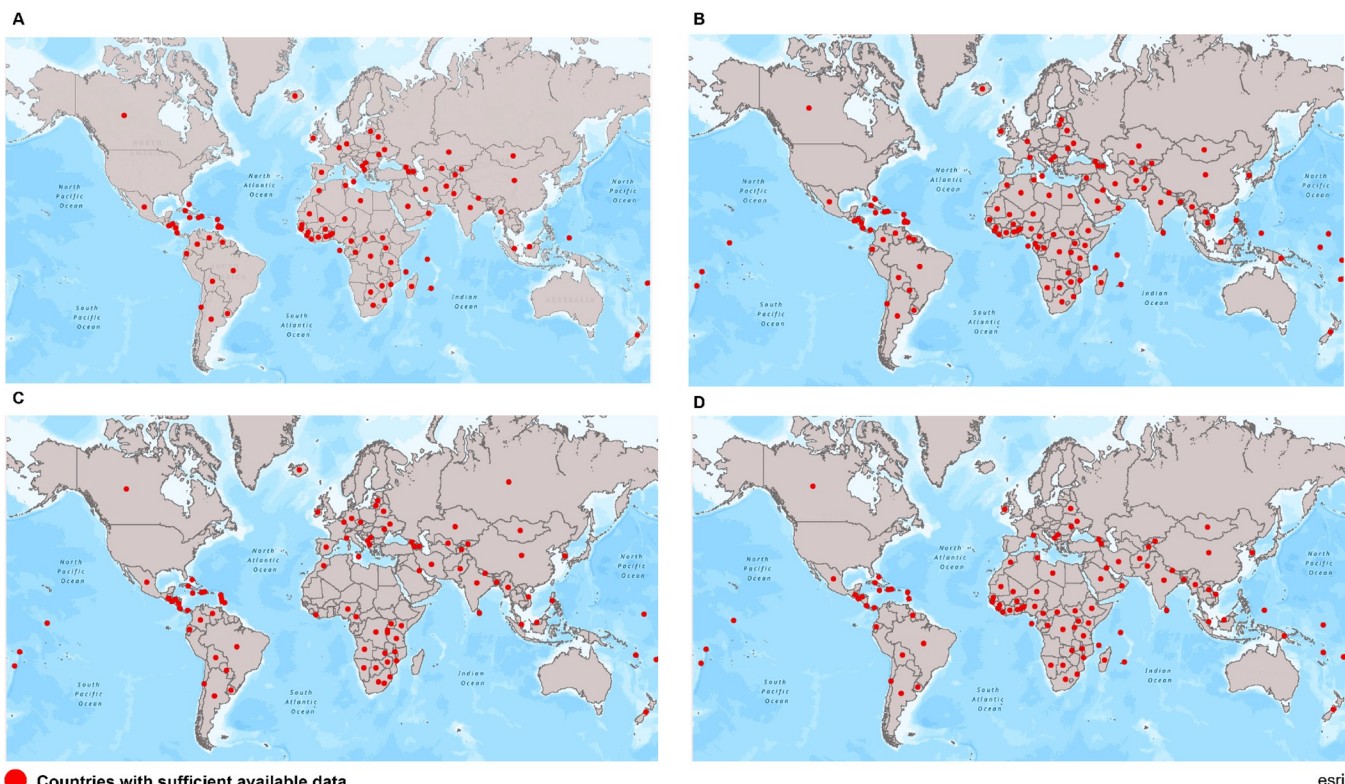

**Fig 3. Countries with sufficient available data for key population stigma indicators. (A)** Sexual behavior/orientation stigma related to men who have sex with men. **(B)** Sex work stigma. **(C)** Gender identity stigma related to transgender persons. **(D)** Drug use stigma. *Data were available for more than 40% of indicators. Base layers of the maps used are from ArcGIS, ESRI: Fig 3A: https://arcg.is/bu8Cu0. Fig 3B: https://arcg.is/Pbzn00. Fig 3C: https://arcg.is/0G0PDG. Fig 3D: https://arcg.is/1HS1zT.

HIV (mathematical) modelers have sought such data on barriers to the AIDS response. For example, currently, we do not know if the people who do not know their HIV status do not know their risk or are avoiding testing due to stigma and discrimination. Several indicators identified in this mapping exercise had been revised in recent years, possibly in an effort to improve reporting and measurement. However, the change in definitions over time limits the comparability of indicators over time and complicates the ability to observe change. Among indicators which do currently have sufficient data to compare time periods, most indicators only have data available for a few time points, and many indicators, particularly at the structural level, necessitate longer intervals and follow-up periods to demonstrate meaningful change. Many of the data obtained in this mapping exercise on laws and policies are not available going back more than a few years, although if continued to be prioritized for collection could be used to track changes in stigma into the future.

Summary measures will be meaningful if they provide an accurate characterization of the overall environment of stigma within countries. Combining the indicators and domains identified in this mapping exercise into fewer items as an index may allow for the creation of summary measures. In order for these summary measures to be concrete and valid, indicators will need to be appropriately aggregated and weighted [50]. Based on the findings of this mapping exercise, there are 2 concurrent approaches that would be appropriate for determining indicator weights. The first is empiric, using methods such as exploratory factor analysis, to assess the underlying factor structure of a given set of variables and the strength of the association between a variable and a latent construct when sufficient data exist [51]. As there are

insufficient data for conducting this process for key population stigma currently, we also propose a second participatory strategy to determine relative importance and weight indicators. Analytic hierarchy process, a multicriteria decision analysis method that seeks to solve a problem through a hierarchical approach [52], is one example of such a participatory approach. A participatory approach will allow for experts from communities, civil society, academia, and public health practices to fill in gaps where data are currently insufficient. Therefore, despite the limitations in the current data available for indicators identified through this mapping exercise, empiric and participatory approaches can be used alongside the data that do exist to inform the creation of summary measures and set data collection priorities moving forward.

There is heterogeneity in the experiences and burden of stigma within and across populations [4,53]. An essential consideration, true in the creation of any summary measure for a complex construct, is that heterogeneity within countries will be masked in the process of distilling multiple indicators into one or even a few indices. This data mapping exercise demonstrated significantly more data characterizing HIV stigmas than those focused on sexual behavior or orientation, gender identity, sex work, or drug use—challenging our ability to describe heterogeneity within the key population subgroups, such as age, residence, employment, etc. Discussions have emerged in this process to consider the development of a single summary measure of stigma for all key populations in order to clearly guide targeted HIV and human rights policy, advocacy, and practice. Not only may this mask diverse experiences of stigma across these populations, but it may also mask substantial differences in the relative state of stigma and progress toward its elimination. For instance, we found vast differences in the level of criminalization related to different behaviors associated with different key populations. However, creating separate indices may artificially segment experiences that are intersectional for individuals with HIV and membership of a key population. Intersectional stigma is the potentially compounded effect of stigmas among individuals with multiple identities which may be devalued by some in society [54]. Unfortunately, the development of measures for intersectional stigma is still in their nascency and would not be represented in this index [27,55]. Additionally, the emerging field of microaggressions is an important area of ongoing research that aims to characterize and understand stigma processes, and, although there is not yet consensus on how this fits within the stigma framework, there is likely great overlap with individual-level experiences of stigmas. However, this is not represented in the current stigma indicators that were considered and whether and how to characterize such experiences at a broader socioecological level to understand stigma at a country level is an important area of future research. Given that summary measures may not be nuanced enough to appropriately represent intersectionality when combined, in balancing all considerations, we recommend the creation of separate indices for HIV and each key population stigmas with ongoing consideration of how to examine intersectional influences on health. The addition of narratives from advocates and people with lived experience will be essential to consider in conjunction with any summary measure toward this purpose.

The process for creating summary measures to accurately describe and track country-level stigma, while also accounting for heterogeneity and complexity is a challenge, yet not an insurmountable one. As described in conceptual frameworks depicting the processes of stigma, these relationships are complex and often do not occur in isolation [17]. In terms of understanding stigma, there is discussion about the appropriate point of measurement—whether it be the social determinants of stigmas (reflected, for example, in indicators such as acceptability of partner violence, demonstration of controlling behaviors, or having experienced intimate partner violence) or the experiences of stigmas as an outcome or endpoint (reflected, for example, in indicators such as having been socially excluded for living with HIV, having experienced HIV discrimination in healthcare, or having lost a source of income due to living with HIV). Continuing this discussion on the process and methods for stigma measurement at a

country level will be an essential step for improving our ability to characterize stigma among and between populations and track country progress in addressing stigma. This should be done in conjunction with, rather than at the expense of, continuing work to measure and assess stigma at other socioecological levels, including the interpersonal and individual levels. Ultimately, measures of stigma at the country level can be used in combination with for instance, individual-level stigma measures, in comparative research to help elucidate the complex interaction between structural stigma and other stigma manifestations.

In addition to the challenges outlined in this discussion, there are several additional limitations to this data mapping exercise that should be considered. The objective of this study was to understand the stigma data that exists for calculating several stigma indicators that have potential use for informing a standardized stigma measure at the country level. As a data mapping exercise, this process was exploratory and therefore may be subject to bias. Although a priori criteria were established and are reported, final decisions on inclusion were based on the state of the data observed during this process. The research and authorship team includes a diverse group of stigma experts from across different academic institutions, multilateral agencies, community networks, and international organizations. This collaborative approach was taken to expand the access to existing stigma data and to minimize selection bias. Our research team used publicly available data, supplemented by data obtained through collaborators, and directly through the research team where possible. This study was therefore unable to assess data that were not publicly available or databases that were not yet available at the time of this data mapping exercise, but could have potentially been shared by institutions outside of this collaborative research team [56,57]. Although systematic reviews of the literature may have informed which data had ever been collected, it would not have yielded the raw data needed to conduct this exercise (i.e., the data would not necessarily be accessible).

## Conclusions

Several indicators were determined to have the potential to appropriately characterize the level and nature of stigma affecting people living with HIV and key populations across countries and time based on the data available. Although few available indicators have sufficient data available to use in summary measures for key populations, empiric and participatory approaches for weighting can be used to fill gaps in the current data and inform more comprehensive summary measures. While many countries lack data for the indicators identified in this data mapping exercise and therefore would be underrepresented in any global characterizations, highlighting these gaps can support the direction of funding opportunities, government endorsement, and supportive technical assistance to certain countries/areas to improve the representativeness of measurement of stigma mitigation progress in the future. The creation of a global stigma data repository would also serve to improve the availability and use of stigma measures, and efforts for increased data collection of validated stigma metrics can improve our ability to characterize country-level stigma across various domains in the future. Using the indicators identified as having sufficient data at present for the creation of country-level indices for HIV stigma, sexual minority stigma related to men who have sex with men, sex work stigma, drug use stigma, and gender identity stigma related to transgender persons can better support tracking progress in stigma mitigation for people living with HIV and key populations.

## Supporting information

**S1 Protocol. Indices on stigma and discrimination related to HIV and key populations (Phase I)**
(DOCX)

**S1 Table. Characteristics of participants in e-consultation online survey.**
(DOCX)

**S2 Table. Proposed indicators for HIV stigma.**
(DOCX)

**S3 Table. Proposed indicators for sexual behavior/orientation stigma related to men who have sex with men.**
(DOCX)

**S4 Table. Proposed indicators for sex work stigma.**
(DOCX)

**S5 Table. Proposed indicators for drug use stigma.**
(DOCX)

**S6 Table. Proposed indicators for gender identity stigma related to transgender persons.**
(DOCX)

## Acknowledgments

We would like to thank all the people living with HIV and members of key populations who agreed to participate in studies across all of the countries where data are available. We would like to thank the community groups and those working to call attention to both measure and address stigmas affecting people living with HIV and key populations. We would like to thank the members of the Task Team developed by UNAIDS Monitoring Technical Advisory Group (MTAG).

The UNAIDS Task Team members not listed as authors in the manuscript were Daniel Reidpath, George Ayala, Erika Castellanos, Ruth Morgan Thomas, Nicholas Niwagaba, Julian Kerboghossian, Sonal Mehta, Joanne Csete, Nazneen Damji, Elena Kudravtseva, Luisa Cabal, Kene Esom, Mianko Ramaroson, and Meaghan Kall.

## Author Contributions

**Conceptualization:** Carrie Lyons, Victoria Bendaud, Taavi Erkkola, Judy Chang, Laura Ferguson, Laura Nyblade, Joseph Amon, Alexandrina Iovita, Eglė Janušonytė, Laurel Sprague, Keith Sabin, Stefan Baral, Sarah M. Murray.

**Data curation:** Carrie Lyons, Victoria Bendaud, Christine Bourey, Ishwarya Ravichandran, Omar Syarif, Pim Looze, Sarah M. Murray.

**Formal analysis:** Carrie Lyons, Christine Bourey, Ishwarya Ravichandran, Sarah M. Murray.

**Methodology:** Carrie Lyons, Victoria Bendaud, Taavi Erkkola, Omar Syarif, Anne Stangl, Laurel Sprague, Stefan Baral, Sarah M. Murray.

**Supervision:** Stefan Baral, Sarah M. Murray.

**Writing – original draft:** Carrie Lyons, Victoria Bendaud, Stefan Baral, Sarah M. Murray.

**Writing – review & editing:** Carrie Lyons, Victoria Bendaud, Christine Bourey, Taavi Erkkola, Ishwarya Ravichandran, Omar Syarif, Anne Stangl, Judy Chang, Laura Ferguson, Laura Nyblade, Joseph Amon, Alexandrina Iovita, Eglė Janušonytė, Pim Looze, Laurel Sprague, Keith Sabin, Stefan Baral, Sarah M. Murray.

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
