## [Editor Report · Decision Letter 0]

24 Jun 2021

Dear Dr Lyons, 

Thank you for submitting your manuscript entitled "Standardizing measures to track progress in eliminating HIV and key population stigma: A data mapping exercise" for consideration by PLOS Medicine.

Your manuscript has now been evaluated by the PLOS Medicine editorial staff and I am writing to let you know that we would like to send your submission out for external assessment.

However, before we can send your manuscript to assessment, we need you to complete your submission by providing the metadata that is required. To this end, please login to Editorial Manager where you will find the paper in the 'Submissions Needing Revisions' folder on your homepage. Please click 'Revise Submission' from the Action Links and complete all additional questions in the submission questionnaire.

Please re-submit your manuscript within two working days, i.e. by Jun 28 2021 11:59PM.

Once your full submission is complete, your paper will undergo a series of checks in preparation for external assessment. 

Kind regards,

Richard Turner, PhD

rturner@plos.org

---

## [Decision Letter · Decision Letter 1]

4 Nov 2021

Dear Dr. Lyons,

Thank you very much for submitting your manuscript "Standardizing measures to track progress in eliminating HIV and key population stigma: A data mapping exercise" (PMEDICINE-D-21-02404R1) for consideration at PLOS Medicine. We apologize for the delay in sending you a response. 

Your paper was discussed with an academic editor with relevant expertise and sent to independent reviewers, including a statistical reviewer. The reviews are appended at the bottom of this email and any accompanying reviewer attachments can be seen via the link below:

[LINK]

In light of these reviews, we will not be able to accept the manuscript for publication in the journal in its current form, but we would like to invite you to submit a revised version that addresses the reviewers' and editors' comments fully. You will appreciate that we cannot make a decision about publication until we have seen the revised manuscript and your response, and we expect to seek re-review by one or more of the reviewers. 

We hope to receive your revised manuscript by Nov 25 2021 11:59PM. Please email us (plosmedicine@plos.org) if you have any questions or concerns.

Please let me know if you have any questions, and we look forward to receiving your revised manuscript. 

Sincerely,

Richard Turner, PhD

Senior editor, PLOS Medicine

rturner@plos.org

Noting PLOS' data policy (https://journals.plos.org/plosmedicine/s/data-availability), please ensure that authors are not named as contacts for inquiries about access to study data in the data statement.

We notice that at least one author is named in the acknowledgements. 

Please mention the dates of data acquisition in the abstract.

Please add a new final sentence to the "Methods and findings" subsection of your abstract, which should begin "Study limitations include ..." or similar, and should quote 2-3 of the study's main limitations.

Please trim the "Conclusions" subsection of the abstract by about 50%.

After the abstract, please add a new and accessible "Author summary" section in non-identical prose. You may find it helpful to consult one or two recent research papers in PLOS Medicine to get a sense of the preferred style. 

At line 95, for example, please remove spaces from within the reference call-outs (i.e., "... HIV pandemic [11,18].".

Please state early in the Methods section (main text) whether the study had a protocol or prespecified analysis plan.

At line 239, please correct "indictors".

Please avoid "leveraged" in favour of "used", for example.

Throughout the text, please substitute "sex" for "gender" where appropriate. 

Please remove the information on funding from the end of the main text. In the event of publication, this information will appear in the article metadata via entries in the submission form. 

Are references 3 & 19 missing journal names?

Please confirm that the maps used in the figures can be published under a CC BY licence. 

Please add a completed checklist for the most appropriate reporting guideline, e.g., STROBE, labelled "S1_STROBE_Checklist" or similar and referred to as such in your Methods section (main text). 

In the checklist, please refer to individual items by section (e.g., "Methods") and paragraph number, not by line or page numbers as these generally change in the event of publication. 

Comments from academic editor:

The reviewers have covered all the issues I have with this submission - the aim of this work is good, and addressing stigma in a methodologically sound manner is important for the reasons outlined in the paper. The reviewers all struggled with the presentation though, and all have provided valuable suggestions to clarify the aim, methods and results. Limitations need to be addressed in the Discussion too.

In the title, it would be helpful to clearly highlight the focus on Stigma (currently that sort of disappears a little amongst the other words).

Given that stigma is likely to be culturally determined, and to vary across populations, possibly even within countries, there should be a clear discussion as to whether there should be a sort of 'basic' stigma data collection that would apply across regions/countries, with some additional summary data to apply to specific countries (which would enable within country tracking of stigma prevalence). Even within country there could be differences by population groups.

Comments from the reviewers:

*** Reviewer #1: 

Alex McConnachie, Statistical Review

The paper by Lyons and colleagues looks at the availability and utility of a range of indicators for measuring stigma relevant to the prevention, diagnosis, and treatment of HIV, derived from a number of international surveys and databases. This review considers the use of statistics in the paper.

Generally, I thought the paper was an interesting read. There is very little for me to review. The only statistics in the paper are those reported in the right-hand columns of Tables 2-6. These could be laid out a little neater. For example, looking at Table 1, indicators 3-6 are difficult to read, as each item goes over two lines, whereas indicators 7-9 have one item per line and are much clearer.

When summarising an indicator that is the percentage of people with a particular response to a survey within a country, it might help to add "%", since this is the unit of measurement. E.g. indicator #1 in Table 1 would be shown as Mean (SD) = 44.4% (20.4%); Range: 5.7% - 81.4%. Note, I think "SD" is better than "sd".

In Table 1, indicator 14 ("Percentage of ever-married women whose husbands/partners demonstrated types of controlling behaviors"), we are given two measures, neither of which matches the definition. However, the second ("proportion of respondents reporting no controlling behaviors") is surely the opposite of what is needed, so could be converted by subtraction from 100%. Also, the summaries given are percentages, not proportions.

For Table 1, indicator 15 ("Percentage of women age 15-49 who have experienced physical and/or sexual violence by an intimate partner in the past 12 months"), two sets of summaries are given, one of which matched the definition (available in 55 countries) and one that is slightly different (available in 53). To what extent do the 55 and 53 countries overlap, or are they different? What does the second definition add? Are the authors saying that it could be used as a proxy? If so, would it be better to report the total number of countries where a value for this indicator (by either definition) is available, and give summaries for the best estimate available, rather than reporting two?

In Table 1, indicator 11, there are two possible sources of information (Civil Society Report, or National Authority Report, though there could be both), but we are given a single set of summaries. For indicator 17, however, we get two sets of summaries, and it is not clear how many countries had data from at least one source. I think it would be better if we could be given a single set of summaries for each indicator, plus an explanation of how data from multiple sources are used to get a single value for each country (e.g. use one source in preference when both available, or use the average of the two).

For the figures, there are a couple of minor observations. Figure 1 is slightly truncated compared to the panels of Figure 2 - it would look better if they were all the same. Also, the relevance of the red and grey dots (or are they asterisks?) was not clear. This also applies to Figure 2, though there are far fewer dots on these.

*** Reviewer #2: 

This study is timely, and very important to ensure that we have a consolidated effort towards addressing stigma. I believe that a key barrier to addressing stigma also lies in the lack of consensus around how stigma can be conceptualized, operationalized, and measured. This study charts a way forward.

I have some comments and suggestions below for your consideration:

Introduction

Overall, some additional references/citations as well as clarity around definitions of stigma would help strengthen the introduction:

1. Lines 77-78: Some citations would be useful to justify this claim

2. Lines 79-85: These are great examples of stigma, however I think as the manuscript is attempting to review measures of stigma, it would be useful for the authors to clarify how anticipated, perceived, internalized, or enacted stigma differ. In fact, enacted (or experienced) stigma often refers to discrimination in most definitions. Some brief definitions for each term introduced and how they might overlap or be distinguished from each other would be useful.

3. Line 100: It would be useful for the reader to define what intersectional stigma refers to.

4. Lines 105-112: A reference/citation for this would be useful for the reader.

Methods

Overall, the methods section had a thick description of the processes involved in the study. Just a few comments and suggestions:

1. Lines 134-135: I think there is sufficient information here on the backgrounds of the participants, but it would be useful to describe if there were any observable differences between the 804 who registered, versus those who eventually participated (if available).

2. Lines 165-166: Would reverse coding refer to index items where questions were reverse coded? Just a brief explanation would help contextualize this better.

3. Line 166: Which summary statistics were calculated for all indicators? It would be useful to mention them here.

4. Lines 167-168: "Indicators were then selected based on their potential for creating a summary measure, also known as an index." - It would be useful to briefly describe how 'potential' was assessed and what processes went into this decision-making stage. Upon further reading throughout the rest of the paragraph, it seems like these were part of the processes to assess 'potential', but it isn't clear if these were done over and above an initial selection process (as described in the first sentence of the paragraph). Some clarity would help,

Results

Overall, the results were detailed and easy to follow. Just some suggestions for the authors' consideration:

1. Line 210: I generally would be cautious of grouping anticipated and experienced stigma together as these are unrelated concepts (i.e. anticipated stigma can be high in the absence of experienced stigma). Some commentary on this in the discussion / limitations could be useful.

2. Lines 271-273: The figures are a great way to illustrate the spread of countries by data availability - however a data table would be useful as supplemental material as well.

Discussion

The discussion is robust and highlights many considerations that the results of this study surface. Some suggestions for the authors' consideration below:

1. Lines 301-306: "These indicators provide some ability for beginning to explore if and how complex experiences of stigma may be able to be summarized to elucidate the state of stigma within a country and assess change going forward." - Can the authors suggest some qualities of these indicators? These would be useful to help guide future participatory approaches to the creation of summary measures.

2. I believe that one key shortcoming in our general discussions around stigma is the lack of consensus around the definitions of stigma. This is evident in the groups of anticipated and experienced stigma - if it is meant to capture individual level stigma, then it should also include internalized stigma. I think future efforts need to involve interdisciplinary academic teams (social psychology, sociology, psychology, health behavior scientists, public health professionals etc.) that have diverged in the ways that they have defined stigma to find consensus over what works for summary indices. Some synonyms include 'felt/perceived' stigma, 'internalized/self' stigma, 'experienced/enacted stigma or discrimination'; some scholars have also argued for stigma such as 'project' stigma, which refers to a form of resistance against stigma.

3. Related to point #2 above, an emerging field where indices are being developed include HIV microaggressions, or microaggressions experienced by key populations. How do microaggressions fit into the overall stigma framework? Overall, I think a paragraph dedicated to addressing the lack of consensus for now, and warning about the divergence in such conceptualizations of stigma, would be important.

*** Reviewer #3: 

GENERAL COMMENTS

The topic the authors address in this manuscript is globally very relevant across different settings. It is very challenging to find a summary measure to routinely quantify the stigma burden related to HIV status or key population membership at national level, which would enable tracking this burden over time in individual countries and comparing it among different countries in a more accessible way. The authors included a number of stakeholders in this exercise, explored an extensive number of data sources and provided an interesting insight into the current status and their view of the way forward. 

However, the lack of clear and detailed description of methods hampers overall clarity of other sections of this work, and it is therefore difficult to judge whether this paper is suitable for publication in Plos Medicine. Several different aims and goals are presented throughout the paper, which is confusing. My understanding is that the authors aimed to review available indicators and data for the period from 2000-2020, track progress over that specific period in individual countries worldwide (retrospectively), summarize the findings in this paper, and them, at some later point, create a summary measure to be used at country level. However, the authors' focus seems to be on the summary measure in the Title/Abstract/Introduction/Discussion, whereas in the Methods/Results they relatively briefly describe their methodological approach. The methods they used seem to have some limitations, substantial bias may have been introduced, a more systematic approach may improve the technical quality. 

It may be useful to present methods in a clearer (step-by-step) way, as it may also be helpful for researchers working on stigma in other areas (outside of HIV) to use it as a roadmap for similar endeavors in their areas in the future.

SPECIFIC COMMENTS (*=major)

Title and Abstract: 

1. The Abstract seems quite long, a shorter, clearer version may be more appropriate to engage the reader 

2. Conclusions in the abstract seem disproportionally long 

3. The title and the abstract are not fully aligned with the content of the main text (e.g. timeframe, inclusion of all countries irrespective of their geographical region/income category etc.)

4. The way the aim of this work is presented in the main text is confusing, the aims in the main text and in the abstract seem inconsistent

Introduction: 

1. The first sentence, L64 is unclear, please rewrite 

2. Suggest using standard abbreviations throughout the text for all key populations, and not only for MSM. Some researchers argue that such abbreviations are dehumanizing, and if this is the reason for authors not to use them, the abbreviations should also be omitted for MSM for consistency. This applies to the Abstract too.

3. Please add reference to support the statement in the first paragraph L77-78. 

4.* Please provide a definition of stigma in the second paragraph. I also suggest a sentence clarifying the terms "measure", "indicator", index" 

5.L103, 104 "A summary measure will allow…" the wording does not reflect the uncertainty, as the measure has not yet been developed; suggest "may allow" or similar; as similar content appears in the last paragraph of this section, consider removing 

6.L105-106 "achieve the achieving", please check and rewrite. There are several other typos throughout the text, please check and correct.

7.* Challenges with the HIV indicators in terms of geographical and temporal consistency are not limited to stigma and discrimination indicators- as this is a journal read by a broad audience, in order not to mislead the readers, authors should present a more balanced perspective of the general challenges with the HIV and key-populations related indicators and M&E. The authors mention progress with the stigma indicators (L97), and that there are still challenges with intersectional stigma and a summary measure - this should be rewritten so that it is clearer for the reader what the main point of the paper actually is, what is the knowledge gap that they are addressing in this paper

8. *In general, there seems to be some repetition throughout this section, it would be easier to follow if it was shorter and some parts, especially those similar to the ones appearing in the Discussion section again removed.

Methods: 

1. L124 "UNAIDS" seems vague, please specify which Team/Office or similar. 

2.* L127 please add reference for GAM. GAM indicators and other established indicators (WHO SI guideline) should be mentioned explicitly in the Introduction, where there's a statement on progress in developing measures (L97)

3.* Several different aims and goals presented in the Introduction section L105-120 and again in the Methods L127-129, this is confusing, please rewrite and present the aims in one paragraph and align the text in the Abstract

4. Suggest to present the whole process (step by step) described in Methods in a flowchart, the text alone is difficult to follow

5.* The way the E-consultations are described in L131-140 is somewhat confusing - it is important for the readers to be able to judge for themselves how balanced and representative the group participating in the e-consultations was, there are no details on what proportion of participants were from which region, stakeholder group etc., a disbalance may have caused bias. If there is a separate report on this process it should be referenced, so that the readers can access it and judge for themselves.

6. Suggest using bullet points for data sources, and adding references to IBBS Studies which were included if they were published or references to their protocols.

7.* IBBS studies implemented by authors' institution and Sub-Saharan Africa only included, which may have introduced bias, the results may therefore not be generalizable in the global context.

8.* The authors did not perform a systematic review of published literature and have not extracted data in a systematic way, with a pre-published protocol available for independent verification, IBBS studies in some countries may have been missed. This is a major methodological limitation, as indicators were later excluded based on the number of countries for which data were available (L169). 

Results:

1.A flowchart with included indicators at every step and excluded indicators (with reasons for exclusion) would add clarity and be easier to follow

2. *Many decisions on exclusion and inclusion of indicators also seem to have been done arbitrary and not in a consistent way (e.g. L185 "Indicators determined to be redundant were removed from consideration" vs L251 "Based on decisions to create separate summary measures for key populations … we did not drop any indicators from further consideration based on potential redundancy.")

3. L273 "the greatest proportion of countries with available data for HIV and key population indicators is in sub-Saharan Africa." is not a surprising finding, as many studies were done there, however, due to the above listed limitations in methodology, some countries in other regions may have been missed. E.g. a systematic review by Fitzgerald-Husek et al. (Ref. 19) covered papers that measured stigma affecting MSM and SW from 2004 to 2014, most of which originated from high income countries (mostly North America), which seems not to be in line with the above statement.

4. Overall, the results are somewhat difficult to follow, please consider visual presentation instead of narrative

Discussion:

1.*This section is difficult to comment as the Methods and Results sections are unclear. The authors, however, do not include their view of the limitations of this study, which would be an important bit.

2. It may be useful to present methods in a clearer (step-by-step) way, as it may be helpful for researchers working on stigma in other areas (outside of HIV) to use it as a roadmap for similar endeavors in their areas in the future and provide comment on relevance for areas outside of HIV too in the Discussion section.

Additional comments:

1.Table 2: I suggest adding a column with data sources where individual indicators were identified, and if there are commonly used or validated indicators in place, which indicators (e.g. GAM, WHO SI etc.)

2.Tables 2-6 Unclear what "Descriptive statistics" actually describes, this should be specified, abbreviation explanations should be added to footnotes of the tables

3.Fig 1 and 2 Footnote-suggest "more" instead of "greater"

*** Reviewer #4: 

The article presents the summary of an interesting exercise that would address a clear need. The nuances of stigma within/between communities and populations would make codifying community-population level indicators (esp at the country-level) challenging, if not nearly impossible. This is indeed true when you factor the various causes of stigma, and the attitudes, policy, and cultural norms that fuel stigma. The scientific rationale for this is clear and established, the methods were sound. There are some grammatical/syntax revisions that should be made. Important consideration for future inquiries about developing summary indicators based on population-level data.

*** Reviewer #5: 

First of all, congratulations for your comprehensive works and great efforts. The manuscript is clearly presented and it was accurately worked by current literatures of HIV to fulfill the study objectives. Sufficient details of methods and analysis are provided and applicable for its interpretation. The findings are described well and the conclusions are drawn adequately supported by the results. 

For more clarification, reviewer' comments are provided as follows:

In methodology, the steps to combine different data for data mapping should be included.

The number of countries extracted from each database should be mentioned for more detailed methodology.

It is interested to know how the denominator is defined for the proportion of countries for which data were available. (Line 190)

It may be better to mention about the reason of setting the cut off of 50% to define sufficient and limited indicators. (Line 192 to 194) 

For "sufficient" available data within the country, is there any consideration of each domain in addition to overall indicators?

In data availability and quality, rationale for exclusion of Violence; Stigma and discrimination; and Internalized stigma for key population should be mentioned like explanation about no consideration of the social Norms and Attitudes domain. (Line 233 to 235)

It is not clear that although it is mentioned "Feeling shame and guilt" and "Avoiding healthcare out of fear of discrimination" were not dropped to be consistent for key population indices (Line 247), these indicators were excluded in table of final indicators for key population because of no data availability. So, it will be better to give other explanation.

It should be mentioned the total number of countries as denominator about availability across countries for HIV stigma and key population stigma. (Line 264)

For figure 1, it may be better to show all UN member states regardless of "sufficient" or "limited" number of HIV stigma indicator like figure 2.

The color legend should be included for clear description of the map for all figures.

Of the 24 potential HIV stigma indicators, it was found that there were 20 indicators that have potential for use in tracking change in stigma over time that shown in the table 2. So please clarify this number that was mentioned as 19 in Line 287. In Line 288, 11 HIV stigma indicators which had data available in more than half of countries were good to mentioned for clear presentation to the readers. It is also better to explain how half of countries is calculated. 

Conclusion in the body of manuscript and in the abstract should be similar.

In line 366, what does "n" mean between limitations and the current data?

It may be better to explain how the domain and subdomain of stigma were considered or defined and how the indicators were categorized for each sub-domain. The literatures based for this categorization should be cited.

***

[LINK]

---

## [Decision Letter · Decision Letter 2]

6 Jan 2022

Dear Dr. Lyons,

Thank you very much for re-submitting your manuscript "Global assessment of existing HIV and key population stigma indicators: a data mapping exercise to inform country level stigma measurement." (PMEDICINE-D-21-02404R2) for consideration at PLOS Medicine.

I have discussed the paper with our academic editor and it was also seen again by four reviewers. I am pleased to tell you that, provided the remaining editorial and production issues are fully dealt with, we expect to be able to accept the paper for publication in the journal.

[LINK]

Please let me know if you have any questions, and we look forward to receiving the revised manuscript.   

Sincerely,

Richard Turner, PhD

rturner@plos.org

Requests from Editors:

In the data statement (submission form), please use the form "... data are ..." throughout. Please check through the main text for consistency on this point too (e.g., "... data exist" at line 149). 

We suggest adding a few words to the abstract to mention the e-consultation and the number of candidate indicators found thereby.

At line 84, rather than "nearly 40 million" please quote an estimate from the relevant reference.

Thank you for your comments about the design of the study. We ask you to include the study protocol as an attachment, labelled "S1_Protocol" or similar and referred to as such in the Methods section, so that readers can judge the extent to which the findings were data-driven. 

We felt that the wording of the first paragraph of the Discussion section (main text) was somewhat more downbeat than the discussion in the abstract and final paragraph of the paper, and you may wish to revisit this component. 

At line 369, we suggest amending the text to "In 2022, ..." or similar. 

Should that be "for key populations" at line 506?

At line 533, should that be "were" rather than "include"?

In the reference list, please amend the typo in reference 2.

Again in reference 18 there is a typo, and it appears that full access details need to be added to this citation. 

Can a report number or URL be added to reference 24?

Please check reference 38 for typos. 

Where available, please add institutional author names to references 39, 40 and others. 

Please use "PLoS Med." and "PLoS ONE" as journal name abbreviations. 

Comments from Reviewers:

*** Reviewer #1: 

Alex McConnachie, Statistical Review

My original comments were mainly to do with tidying up the presentation, and the authors have adequately responded to all of them. I have no further comments to make.

*** Reviewer #2: 

Dear authors,

Thank you for the opportunity to review your resubmission of the present manuscript. Appreciate the thoughtful and detailed responses raised to the initial comments provided. Overall, all major and minor comments have been addressed satisfactorily. Additionally, the authors also sufficiently address and discuss the complexities and nuances in the extant stigma literature, which have allowed for the present work to be clearly situated and therefore understood in often diverging definitions/approaches to stigma. The scope of the work is also clarified in this iteration of the manuscript. I only have a few minor comments left at this point:

Line 103-104: "Perceived stigmas refer to felt experiences of stigma or discrimination and the perception of bias4 as understood by a person living with a stigmatized identity" - I appreciate the clarifications made by the authors in defining all the forms of stigma in the revision. I would be careful with the use of the word "experiences" here as it may be confused with "Experienced Stigma". Furthermore, 'felt... discrimination' would also be somewhat confusing because discrimination already implies an action that has been done unto the stigmatized individual. 

Overall, removing "...experiences of..." and "or discrimination" from this sentence would suffice.

Line 106: "... discriminatory acts by someone." - perhaps adding something to the effect of "...on the basis of a stigmatized identity." would help clarify this.

Line 143: "interpretating"  "interpreting"

*** Reviewer #3: 

The authors provided detailed point-by-point responses. They substantially improved the overall clarity of their paper, and in particular, the clarity of the Methods section. They also strengthened the paper by providing an extensive overview of the limitations in the Discussion section, and outlined the opportunities for improvement in the future in a more straightforward way. The alignment of different parts of their manuscript seems much better in this version.

*** Reviewer #5: 

Dear Author, 

Thank you for your each and every response to my comments of the first review. The second version is satisfied for further proceedings but some minor revisions are suggested as the followings:

For Table 2, 

(1) There is sign of " 3 asterisks" in the foot note but the data or text referred was not found within the table.

(2) The indicator of sub-Domain "Discrimination anticipated in health care settings" is needed to be numbered as "19" and the domain "Anticipated stigma" is good to be mentioned before the "Experienced stigma" domain. Therefore the order of three indicators under "Experienced stigma" domain will be 20,21 and 22.

In the line 346, the asterisk included in the tile of Fig 3 should be removed because the color legend has shown for the figure. 

For Fig 3, the color legend as "Countries with sufficient available data" may be better to be complete.

Best Regards,

May Soe Aung

***

[LINK]

---

## [Editor Report · Decision Letter 3]

14 Jan 2022

Dear Dr Lyons, 

On behalf of my colleagues and the Academic Editor, Dr Newell, I am pleased to inform you that we have agreed to publish your manuscript "Global assessment of existing HIV and key population stigma indicators: a data mapping exercise to inform country level stigma measurement." (PMEDICINE-D-21-02404R3) in PLOS Medicine.

Prior to final acceptance, please:

Add an additional sentence, say, to the "Methods and findings" subsection of your abstract, we suggest at line 53, to describe the indicators in general terms (e.g., "These indicators should allow assessment of legal, societal and behavioural manifestations of stigma across population groups and settings."); and 

Split the final point of the "Author summary" into two, at "however".

PRESS

Sincerely, 

Richard Turner, PhD 

rturner@plos.org